# Treatment of *Helicobacter pylori* Infection in Patients with Penicillin Allergy

**DOI:** 10.3390/antibiotics12040737

**Published:** 2023-04-10

**Authors:** Ligang Liu, Milap C. Nahata

**Affiliations:** 1Institute of Therapeutic Innovations and Outcomes (ITIO), College of Pharmacy, The Ohio State University, Columbus, OH 43210, USA; liu.10645@osu.edu; 2College of Medicine, The Ohio State University, Columbus, OH 43210, USA

**Keywords:** penicillin allergy, *Helicobacter pylori*, vonoprazan, bismuth quadruple therapy

## Abstract

*Helicobacter pylori* is among the prevalent causes of infections worldwide, and its resistance rate to antibiotics has been rising over time. Amoxicillin is the cornerstone for the treatment regimen. However, the prevalence of penicillin allergy ranges from 4% to 15%. In patients with true allergy, Vonoprazan-Clarithromycin-Metronidazole and bismuth quadruple therapy have demonstrated excellent eradication and high adherence rates. Vonoprazan-based therapy is administered less frequently and may be better tolerated than bismuth quadruple therapy. Therefore, vonoprazan-based therapy may be considered as a first-line therapy if accessible. Bismuth quadruple therapy can be used as the initial therapy when vonoprazan is unavailable. Levofloxacin or sitafloxacin-based regimens achieve a moderately high eradication rate. However, these are associated with potentially serious adverse effects and should only be used when other effective and safer regimens are unavailable. Cephalosporins such as cefuroxime have been used as an alternative to amoxicillin. Microbial susceptibility studies can guide the selection of appropriate antibiotics. PPI-Clarithromycin-Metronidazole fails to achieve a high eradication rate and should be used as a second-line therapy. PPI-Clarithromycin-Rifabutin should not be used because of low eradication rate and frequent adverse reactions. The choice of the most effective antibiotic regimen can enhance clinical outcomes in patients with *H. pylori* infection and penicillin allergy.

## 1. Introduction

*Helicobacter pylori* infection continues to be the most common infection, affecting approximately 44% of the population worldwide, with prevalence varying by region and socioeconomic status [1]. It is strongly associated with peptic ulcer disease, functional dyspepsia, and even gastric cancer and other gastrointestinal malignancies [2,3]. Moreover, it is also related to extra-gastroduodenal diseases, including metabolic disorders, cardiovascular disease, cerebrovascular disease, neurodegenerative disease, and other conditions [4].

The treatment of *H. pylori* infection has faced many challenges due to increasing antibiotic resistance [5]. Resistance rates for clarithromycin range from 17% to 31.5%, levofloxacin from 15.8% to 37.6%, and metronidazole from 38.9% to 44%, with variations depending on the geographic region [6,7,8]. However, amoxicillin had a low resistance rate of less than 3% to 5% in most countries [9]. Amoxicillin is an essential component of the regimen for the eradication of *H. pylori* due to its low resistance rate, safety, affordability, and availability [10,11,12,13]. It is important to note that 5% to 15% of patients in developed countries have reported a penicillin allergy, with rates of 9.3% in South Australia and 10% in the US [14,15,16]. In China, Hong Kong, and Japan, the reported rates of penicillin allergy ranged from 4% to 5.6% [17]. 

Both the American College of Gastroenterology (ACG) guideline and the Toronto Consensus recommend bismuth quadruple therapy as a treatment for patients with penicillin allergy [10,12]. Similarly, the Maastricht VI/Florence consensus report also suggested that bismuth quadruple therapy was the preferred treatment option for this population [18]. Bismuth is not available as a medication in Japan. The Japanese guidelines recommend proton pump inhibitor (PPI), clarithromycin, and metronidazole regimen if susceptible to clarithromycin [13]. Importantly, the Chinese Consensus Report on *Helicobacter pylori* infection does not make recommendations for the treatment [11]. Recent studies have shown that the vonoprazan-based regimen was at least as effective as the PPI-based regimen [19]. In addition, several studies have reported promising results with vonoprazan containing antibiotic regimen in patients with penicillin allergy [20,21,22]. 

The objective of this review is to report the efficacy, safety, and adherence of various treatments available for *H. pylori* infection in patients with confirmed penicillin allergy, based on evidence gathered from clinical trials and real-world observational studies.

## 2. Literature Search

PubMed, EMBASE, and SCOPUS databases were searched, using the keywords: allergy, anaphylaxis, hypersensitivities, allergies, “allergic reaction”, “allergic reactions”, or allergic, *Helicobacter*, “*Helicobacter pylori*”, “*H. pylori*”, penicillin, beta-lactam, b-Lactam in reception to 18 January 2023. A total of 22 studies were included in this review. 

## 3. Evidence for the Treatment of *H. pylori* Infection in Patients with Penicillin Allergy

The characteristics of the included studies are summarized in Table 1. It is important to note that the majority of studies conducted in patients with penicillin allergy included a relatively small number of subjects and were often retrospective, uncontrolled, or single cohort in study design. All results of eradication rate are presented on both an intention-to-treat and a per-protocol basis. The eradication rates, incidence of adverse events, and adherence to vonoprazan-based therapy, PPI-based triple therapy, and bismuth quadruple therapy are presented in Table 2, Table 3, and Table 4, individually.

**Table 1 antibiotics-12-00737-t001:** Characteristics of included studies.

Authors (Year, Country)	Study Design	Patient Characteristics	Treatment Regimens and Confirmatory Test for *H. pylori* Eradication
Zhang et al. (2022, China) [23]	RCT	Age: 18–75 years Penicillin allergy: not well defined Without previous eradication therapy Peptic ulcer: 15.8–19.4%	PBM1M2: R 10 mg BID + M2 100 mg BID + M1 400 mg TID + B 220 mg BID PBLM2: R 10 mg BID + M2 100 mg BID + L 400 mg QD + B 220 mg BID Duration: 14 days Confirmed: UBT at 4–12 weeks after treatment
Tepes et al. (2021, Slovenia) [24]	Analysis of multicenter prospective registry	Age: 18–90 years Penicillin allergy: not well defined Without previous eradication therapy Peptic ulcer: N/A	PC1M1: E 40 mg BID + C1 500 mg BID + M1 400 mg TID Duration: 14 days Confirmed: UBT at least 4 weeks after treatment.
Sue et al. (2021, Japan) [22]	Single-center, open-label, single-arm	Age: ≥20 years Penicillin allergy: diagnosis by a physician Failed first-line therapy Gastroduodenal ulcer: 23.5%	VM1S: V 20 mg BID + M1 250 mg BID + SF 100 mg BID Duration: 7 days Confirmed: UBT at 4 weeks after treatment
Nyssen et al. (2020, European) [25]	Prospective registry of the clinical practice of European gastroenterologists (27 countries)	Age: Mean 53 ± 15 years Allergic to penicillin: not well defined With or without prior H pylori treatment Peptic ulcer: 17%	PC1M1: PPI + C1 + M PC1L: PPI + C1 + L PBTM1: PPI + B + T + M PM1L: PPI + M + L PC1LM1: PPI + C1 + L+ M Duration: N/A Confirmed: Locally accepted/validated diagnostic methods at least 4 weeks after treatment
Gao et al. (2019, China) [26]	Retrospective	Age: 19–75 years Penicillin allergy: not well defined Without previous eradication therapy Peptic ulcer: 17.5%	PBTM1: LPZ 30 mg BID + B 150 mg TID + T 500 mg TID + M1 400 mg TID Duration: 14 days Confirmed: UBT at 4 weeks after treatment
Song et al. (2019, China) [27]	Prospective single center	Age: 42.8 ± 13.7 years Penicillin allergy: well defined Without previous eradication therapy Peptic ulcer: 9.2%	Cefuroxime 500 mg BID + L 500 mg QD + E 20 mg BID + B 220 mg BID Duration: 14 days Confirmed: UBT at 8–12 weeks after treatment
Long et al. (2018, China) [28]	Prospective, randomized, open-label, single-center	Age: 25–65 years Penicillin allergy: not well defined Without previous eradication therapy Peptic ulcer: 15.2%	PC1M1: E 20 mg BID + C1 500 mg BID + M1 400 mg QID BEC1M1: E 20 mg BID + C1 500 mg BID + M1 400 mg QID + B 600 mg BID Duration: 14 days Confirmed: UBT at 6 weeks after treatment
Osumi et al. (2017, Japan) [29]	Retrospective	Age: 26–83 years Penicillin allergy: not well defined Unknown prior treatments Peptic ulcer: N/A	20 mg R + 250 mg M1 + 100 mg M2 BID Duration: 7 days Confirmed: UBT at least 12 weeks after treatment
Sue et al. (2017, Japan) [30]	Prospective study for vonoprazan, Retrospective for PPI	Age: ≥20 years Penicillin allergy: not well defined Without previous eradication therapy Peptic ulcer: 28%	VC1M1: V 20 mg + C1 200 or 400 mg + M1 250 mg BID PC1M1: PPI (30 mg BID LPZ or 20 mg BID E) + C1 200 or 400 mg BID + M1 750 mg BID Duration: 7 days Confirmed: UBT at least 4 weeks after treatment
Mori et al. (2017, Japan) [31]	Prospective, single arm, nonrandomized	Age: ≥20 years Penicillin allergy: not well defined With or without prior treatment Peptic ulcer: 15.8%	PM1S: 20 mg E BID + 250 mg M1 BID + 100 mg SF BID Duration: 10 days Confirmed: UBT or HpSA at 12 weeks after treatment
Ono et al. (2017, Japan) [32]	Retrospective	Age: mean 59 years Penicillin allergy: not well defined With or without prior treatment Peptic ulcer: 17.8%	PC1M1: C1 200 mg BID + M1 250 mg BID + LPZ 30 mg BID or R 20 mg BID VC1M1: C1 200 mg BID + M1 250 mg BID + V 20 mg BID PM1S: M1 250 mg BID + SF 100 mg BID + LPZ 30 mg BID or R 20 mg BID VSM1: M1 250 mg BID + SF 100 mg BID + V 20 mg BID Duration: 7 days Confirmed: UBT more than 8 weeks after treatment
Segarra-Newnham et al. (2004, USA) [33]	Retrospective	Age: 29–84 years Penicillin allergy: not well defined Unknown prior treatments Peptic ulcer: N/A	PC1M1: M1 500 mg BID + C1 500 mg BID + PPI BID Duration: 7 days Confirmed: No information
Parch et al. (1998, USA) [34]	Single-center, open, randomized, parallel	Age: 18–75 years Penicillin allergy: not well defined Unknown prior treatments Peptic ulcer: 100%	O 20 mg BID + C1 500 mg TID Duration: 7 days Confirmed: UBT at 4 weeks after treatment
Gisbert et al. (2010, Spain) [35]	Prospective multicenter	Age: mean 51 ± 18 years Penicillin allergy: not well defined Without prior treatment, start PC1M1 first, if failed, start PC1L Peptic ulcer: 98%	PC1M1: O 20 mg BID + C1 500 mg BID + M1 500 mg BID for 7 days PC1L: O 20 mg BID + C1 500 mg + L 500 mg BID for 10 days. Duration: 7 days vs. 10 days Confirmed: UBT at 8 weeks after treatment
Tay et al. (2012, Australia) [36]	Retrospective	Age: mean 16–85 years Penicillin allergy: not well defined Failed prior treatment Peptic ulcer: N/A	R 20 mg TID + B 240 mg QID + RFB 150 mg BID + C2 500 mg BID Duration: 10 days Confirmed: UBT at least 4 weeks after treatment
Liang et al. (2013, China) [37]	Prospective, open label	Age: 18–79 years Penicillin allergy: not well defined Failed one or more prior treatments Peptic ulcer: N/A	N = 109, results reported for whole group including non-allergic PBTF: LPZ 30 mg BID + B 220 mg BID + T 500 mg TID + FZD 100 mg TID PBTM1: LPZ 30 mg BID + B 220 mg BID + T 500 mg QID + M1 400 mg QID Duration: 14 days Confirmed: UBT at 6 weeks after treatment
Matsushima et al. (2006, Japan) [38]	Retrospective	Age: ≥18 years Penicillin allergy: not well defined Without prior treatment Peptic ulcer: N/A	PTM1: PPI (LPZ 30 mg, O 20 mg, or R 10 mg, QD) + T 500 mg BID + M1 250 mg BID Duration: 7–14 days Confirmed: UBT or by HP stool antigen at more than 2 months after therapy
Gisbert et al. (2005, Spain) [39]	Prospective single center	Age: ≥18 years Penicillin allergy: not well defined With or without prior treatment Peptic ulcer: N/A	PC1M1: O 20 mg BID + C1 500 mg BID + M1 500 mg BID for 7 days BTM1: ranitidine bismuth citrate 400 mg BID + T 500 mg QID + M1 250 mg QID for 7 days PCR: RFB 150 mg BID + C1 500 mg BID + 3rd line O 20 mg BID for 10 days PC1L: L 500 mg BID + C1 500 mg BID + O 20 mg BID for 10 days Duration: 7 days or 10 days Confirmed: UBT at 8 weeks after treatment
Rodriguez-torres et al. (2005, Puerto Rico) [40]	Prospective	Age: ≥21 years Penicillin allergy: not well defined With or without prior treatment Peptic ulcer: 0%	PM1T: E 40 mg + T 500 mg + M1 500 mg QID Duration: 14 days Confirmed: UBT at 4 weeks after treatment
Gisbert et al. (2015, Spain) [41]	Prospective multicenter	Age: ≥21 years Penicillin allergy: not well defined With or without prior treatment Peptic ulcer: 9%	PC1M1: O 20 mg + C1 500 mg + M1 500 mg BID for 7 days. PBTM1: O 20 mg BID + B 120 mg QID + Oxytetracycline 500 mg QID or doxycycline 100 mg BID + M1 500 mg TID for 10 days PC1L: O 20 mg + C1 500 mg + L 500 mg BID for 10 days PC1R: O 20 mg BID + C1 500 mg BID + RFB 150 mg BID for 10 days Duration: 7–10 days Confirmed: UBT at 8 weeks after treatment
Adachi et al. (2023, Japan) [20]	Retrospective	Age: > 18 years Penicillin allergy: self-reported With or without prior treatment Peptic ulcer: 3.8%	PC1M1: C1 200 mg BID + M1 250 mg BID + L 30 mg BID or R 20 mg BID or E 20 mg BID VC1M1: C1 200 mg BID + M1 250 mg BID + V 20 mg BID VM1S: M1 250 mg BID + STFX 50 mg BID + V 20 mg BID Duration: 7 days Confirmed: UBT 6 to 8 weeks after treatment
Gao et al. (2023, China) [21]	Cross-sectional retrospective	Age: 18–69 years Penicillin allergy: well defined Without prior treatment Peptic ulcer: 33.8%	V 20 mg BID + T 500 mg TID (body weight < 70 kg) or QID (body weigh ≥ 70 kg) Duration: 14 days Confirmed: UBT at least 6 weeks after therapy

Abbreviation: RCT, randomized controlled trial; UBT, urea breath test; BID, twice daily; TID, three times daily; QID, four times daily; QD, one times daily; PP, per protocol analysis; ITT, intention to treat analysis; B, Bismuth compound; C1, Clarithromycin; C2, Ciprofloxacin; E, Esomeprazole FZD, Furazolidone; L, Levofloxacin; M1, Metronidazole; M2, minocycline O, Omeprazole; PPI, proton pump inhibitor; R, Rabeprazole; RBC, ranitidine bismuth subcitrate; RFB, Rifabutin; SF, Sitafloxacin; T, Tetracycline; V, Vonoprazan; LPZ, lansoprazole; HpSA, Helicobacter pylori stool antigen; PM1L, PPI-Metronidazole-Levofloxacin; PC1L, PPI-Clarithromycin-Levofloxacin; PC1M1, PPI-Clarithromycin-Metronidazole; PM1T, PPI-Metronidazole-Tetracycline; PC1R, PPI-Clarithromycin-Rifabutin; PC1M1L, PPI-Clarithromycin-Metronidazole-Levofloxacin; PBTM1, PPI-Bismuth-Tetracycline-Metronidazole; N/A, not available; PBCM1, PPI-Bismuth-Clarithromycin-Metronidazole; PBRC2, PPI-Bismuth-Rifabutin-Ciprofloxacin; PBTF, PPI-Bismuth-Tetracycline-Furazolidone; PBM1M2, PPI-Bismuth-Metronidazole-Minocycline; PBLM2, PPI-Bismuth-Levofloxacin-Minocycline; VSM1, Vonoprazan-Sitafloxacin-Metronidazole; VC1M1, Vonoprazan-Clarithromycin-Metronidazole; PM1S, PPI-Metronidazole-Sitafloxacin.

### 3.1. Vonoprazan-Based Therapy 

#### 3.1.1. Vonoprazan-Clarithromycin-Metronidazole (VC1M1) Therapy

Efficacy: 

First-line therapy: Sue et al. [30] found the combination therapy of vonoprazan, clarithromycin, and metronidazole (VC1M1) for 7 days achieved an eradication rate of 100% in a prospective study. Similarly, Ono et al. [32] reported a high eradication rate of 92.3% for 7-day VC1M1 therapy. 

Second-line or third-line therapy: Ono et al. [32] used the same regimen, and successful eradication was observed in one patient who failed initial therapies. This finding suggests that VC1M1 may be effective in patients who have failed other treatments, although further research is needed to confirm the efficacy of this regimen under this situation.

Unknown-line therapy: In a retrospective study, a 7-day course of VC1M1 was used to treat patients who had penicillin allergy with or without previous therapies [20]. This combination achieved a high eradication rate of 94.3% [20]. Additionally, in patients with clarithromycin-resistant infection, the eradication rate was 90.9%. These findings suggest that vonoprazan-based therapy is an effective treatment for *H. pylori* infection in patients with penicillin allergy, including those with clarithromycin-resistant infections. 

Safety: 

Overall, the VC1M1 regimen was safe, with mild and tolerable adverse effects such as diarrhea, abdominal discomfort, fatigue, and headache [20,30,32]. The medication adherence rate was high, ranging from 97.1% to 100% [20,30,32]. However, one patient discontinued treatment due to edema [20].

#### 3.1.2. Vonoprazan-Sitafloxacin-Metronidazole (VSM1) Therapy

Efficacy:

First-line therapy: Ono et al. [32] found that the combination of vonoprazan, sitafloxacin, and metronidazole (VSM1) for 7 days achieved a 100% eradication rate in a retrospective study. 

Second-line or later-line therapy: The eradication rate of 7-day VSM1 was 88.2% in patients who had not responded to initial treatment [22]. For three patients who previously failed at least one therapy, VSM1 achieved an eradication rate of 66.7% [32]. In 10 patients with penicillin allergy with or without previous therapies, the eradication rate was 90% [20].

Safety:

The overall adherence to this regimen was excellent, with a rate of 100% [20,22,32]. The adverse reactions were mild, including diarrhea, abdominal fullness, and abdominal pain [20,22,32]. However, one patient experienced a severe adverse event in the form of skin eruption or bloody stools after receiving this regimen [32].

#### 3.1.3. Vonoprazan-Tetracycline Therapy

A recent study by Gao et al. [21] investigated the efficacy of a new combination therapy consisting of vonoprazan and tetracycline. The study found this regimen achieved a 100% eradication rate in treatment-naive patients. The adherence rate was 94.4%. Adverse events were well tolerated, with patients recovering spontaneously after discontinuing therapy.

**Table 2 antibiotics-12-00737-t002:** Results of vonoprazan-based regimens for Helicobacter pylori eradication therapy.

Authors (Year, Country)	Treatment Status	N	Eradication Rate		Adverse Events	Adherence
			ITT	PP		
Vonoprazan-Clarithromycin-Metronidazole (VC1M1)						
Sue et al. (2017, Japan) [30]	First-line	20	100%	100%	Diarrhea (5.0%) Nausea (15%) Abdominal pain (15%) Abdominal fullness (30%) General malaise (15%) Headache (10%)	100%
Ono et al. (2017, Japan) [32]	First-line	13	92.3%	92.3%	Severe adverse events (0%)	N/A
Ono et al. (2017, Japan) [32]	Second-line	1	100%	100%	Severe adverse events (0%)	N/A
Adachi et al. (2023, Japan) [20]	Not clear	35	94.3% CAM-R (90%) CAM-S (100%)	100% CAM-R (100%) CAM-S (100%)	8.6%	97.1%
Vonoprazan-Sitafloxacin-Metronidazole (VSM1)						
Ono et al. (2017, Japan) [32]	First-line	14	92.9%	100%	Severe adverse events (5.9%)	N/A
Ono et al. (2017, Japan) [32]	Second-line	3	66.7%	66.7%		N/A
Adachi et al. (2023, Japan) [20]	Not clear	10	90%	90%	20%	100%
Sue et al. (2021, Japan) [22]	Second- or later-line	17	88.2%	88.2%	Diarrhea (50.0%) Dysgeusia (6.3%) Nausea (6.3%) Abdominal pain (31.3%) Headache (12.5%) Abdominal fullness (50.0%) General malaise (12.5%) Hives (25.0%) Belching (25.0%)	100%
Vonoprazan-Tetracycline (VT)						
Gao et al. (2023, China) [21]	First-line	18	100%	N/A	Total (27.8%) Nausea (5.6%) Skin rash (5.6%) Fatigue (5.6%) Abdominal pain (11.2%)	94.40%

Abbreviation: N, number of patients; PP, per protocol analysis; ITT, intention to treat analysis; CAM-S, clarithromycin-sensitive; CAM-R, clarithromycin-resistant; N/A, not available.

### 3.2. PPI-Based Therapies 

#### 3.2.1. PPI Clarithromycin Dual Therapy 

Prach et al. [34] conducted a study using omeprazole and clarithromycin to treat *H. pylori* in patients with penicillin allergy in 1998, reporting a 100% eradication rate in three patients. However, the current surge in clarithromycin resistance has decreased the effectiveness of clarithromycin. Therefore, a combination of two antibiotics with PPI has been recommended to achieve high eradication rates.

#### 3.2.2. PPI-Based Triple Therapy 

##### PPI-Clarithromycin-Metronidazole (PC1M1) Therapy

Several studies have explored the efficacy and safety of PPI-Clarithromycin-Metronidazole for patients with penicillin allergy [24,25,28,30,32,35,39,41]. These studies utilized different PPIs, such as omeprazole, lansoprazole, esomeprazole, and rabeprazole, and the dose of metronidazole ranged from 500 mg to 1600 mg daily. The studies carried out in Japan used 200 mg clarithromycin, while those conducted in other countries used 500 mg clarithromycin twice daily. 

Efficacy:

First-line therapy: Ono et al. [32] reported a low eradication rate of 55.6% with a 7-day standard dose PPI in combination with metronidazole 250 mg, and clarithromycin 200 mg twice daily. Gisbert et al. [35,39,41] conducted three prospective studies using omeprazole 20 mg, metronidazole 500 mg, and clarithromycin 500 mg twice daily for 7 days. The eradication rates ranged from 59% to 64% in treatment-naïve patients. Nyssen et al. [25] analyzed data from the European Registry on *H. pylori* management and reported an eradication rate of 69% for first-line therapy. However, this study did not provide information on dosing and duration of therapy [25].

Two studies used metronidazole 1500 mg to 1600 mg daily instead of 1000 mg daily in treatment-naive patients [28,30]. Long et al. [28] observed that PPI, clarithromycin, and metronidazole 400 mg four times daily for 14 days reached an eradication rate of 70%. A 7-day PPI, clarithromycin, and metronidazole 750 mg twice daily achieved a higher eradication rate of 82.7% [30]. Metronidazole is a concentration-dependent antibiotic; thus, higher individual doses each day may have increased the eradication rate. A study demonstrated that the eradication rate of PPI-Clarithromycin-Metronidazole was higher when it was administered for 14 days compared to 7 days, with an eradication rate of 83% [24].

Second-line therapy: Ono et al. [32] reported a low eradication rate of 33.3% for second-line therapy in three patients who had failed initial therapy.

Lack of knowledge on previous therapies: For patients with unknown previous therapy status, Adachi et al. [20] stated that 7-day PC1M1 achieved an eradication rate of 50%. Segarra-Newnham et al. [33] reported a 91% eradication rate in veterans treated with a combination of PPI, metronidazole 500 mg, and clarithromycin 500 mg twice daily between 1996 and 2001. However, with the alarming antibiotic resistance, PC1M1 may not achieve eradication rates as high as 90%.

Safety:

Based on the evidence from these studies, a lower dose of metronidazole was associated with a lower incidence of adverse events ranging from 7.7% to 17%, and the most common adverse effects included nausea and diarrhea [32,35,39,41]. The medication adherence rate for this lower dose regimen was generally high, ranging from 92% to 98% [32,35,39,41]. On the other hand, using a higher dose of metronidazole was associated with a higher incidence of adverse events, at 45.5% [28]. Most of these adverse events were well-tolerated, such as nausea, abdominal pain, and taste distortion. The adherence rate for the higher dose was 93.9%. Taste distortion most likely occurred with the high dose of metronidazole [42].

#### 3.2.3. PPI-Metronidazole-Tetracycline (PM1T) Therapy

Efficacy: 

First-line therapy: PM1T achieved relatively high eradication rates among penicillin-allergic patients who did not receive any previous therapies [38,40]. Rodriguez-Torres et al. [40] reported the eradication rate was 84% with PM1T, while Matsushima et al. [38] found the eradication rate was 80% among five patients with penicillin allergy. 

Second-line therapy: PM1T achieved an eradication rate of 100% in three patients as a second-line therapy [40]. However, due to the small sample size, it would not be appropriate to conclude that it should be used as a second-line treatment. Larger studies would be needed to confirm these results and to determine the generalizability to a broader patient population.

Safety: 

Despite the high eradication rate observed with PM1T, a large proportion of patients experienced mild to moderate adverse events. In one study, 55% of patients experienced adverse events, while 20% of patients discontinued treatment due to oral candidiasis and diarrhea [40]. In another study involving five patients, no severe adverse events were reported, but some of them experienced taste disturbance and soft stools [38]. 

#### 3.2.4. PCL: PPI-Clarithromycin-Levofloxacin (PC1L) Therapy

Efficacy: 

First-line therapy: The eradication rate of PC1L was 82% in patients who had not previously received treatment for Helicobacter pylori infection [25]. 

Second-line or later-line therapy: For patients who failed PC1M as the first-line therapy, the eradication rate with PC1L was between 69% and 73% [25,35,41]. The eradication rate was 64% in patients who failed PBTM1 [41]. PC1L achieved a 100% success rate in three patients in another study [25]. For patients who failed both PC1M1 and PBTM1, the eradication rate with PC1L was 50% in five patients [25,41]. PC1L achieved a high eradication rate of 100% in patients who failed three regimens [39,41]. 

Safety: 

In studies with a relatively large sample size, the incidence of adverse effects ranged from 16% to 29% when used as a first-line or second-line therapy [25,35,41]. The adherence rate was between 88% and 100% [25,35,41]. In seven patients with multiple treatment failures, the incidence of adverse events was over 50%. The adherence ranged from 67% to 100% [39,41]. 

#### 3.2.5. PPI-Metronidazole-Levofloxacin (PM1L) Therapy

Only one study reported the efficacy and safety of PM1L for the treatment of *H. pylori* as a second-line therapy [25]. PM1L achieved an eradication rate of 77% in 13 patients who failed clarithromycin-based triple therapy. Among them, 23% of patients experienced adverse events, and none discontinued treatment. For four patients with failure to bismuth quadruple therapy (PBTM1), the eradication rate with PM1L was 75%, no adverse effects were reported, and the adherence rate was 100%. 

#### 3.2.6. PPI-Metronidazole-Sitafloxacin (PM1S) Therapy

Efficacy: 

In treatment-naïve patients, two studies found the eradication rate of PM1S reached 100% [31,32]. Mori et al. [31] reported an eradication rate of 84.2% in patients who failed one regimen, and 40% in five patients who failed two courses of therapies. The eradication rate was 100% when it was used as a second-line or third-line therapy in 24 patients [32]. 

Safety: 

Mori et al. [31] reported that 31.6% of patients experienced mild, tolerable adverse events, and the medication adherence was 100%. Ono et al. [32] found the incidence of severe adverse events was 2.3%, and only one patient discontinued therapy. 

#### 3.2.7. PPI-Clarithromycin-Rifabutin (PC1R) Therapy

Efficacy: 

Two prospective studies explored the safety and efficacy of 10-day omeprazole 20 mg, clarithromycin 500 mg, and rifabutin 150 mg twice daily [39,41]. The eradication ranged from 17% to 20% when it is used as a third-line therapy. PC1L did not eradicate infection as a fourth-line therapy in two patients who failed PC1M1, PBTM1, and PCL [41].

Safety: 

As a third-line therapy, adverse events occurred in 71% to 89% patients, causing over 60% of patients to discontinue treatment [39,41]. All patients experienced adverse events and stopped treatment when it was used as a fourth-line therapy [41].

#### 3.2.8. PPI-Metronidazole-Minocycline (PM1M2) Therapy

Osumi et al. [29] observed an eradication rate of 100% using 7-day 20 mg rabeprazole, 250 mg metronidazole, and 100 mg minocycline twice daily in five patients. However, the study did not provide any details regarding adverse events or adherence during the treatment period.

### 3.3. PPI-Clarithromycin-Metronidazole-Levofloxacin (PC1M1L) Therapy

In one patient who failed traditional bismuth quadruple therapy and PPI-Metronidazole-Levofloxacin regimen, a successful eradication of *H. pylori* by PC1M1L was observed [25]. No adverse events were observed, and this patient was adherent to the treatment.

**Table 3 antibiotics-12-00737-t003:** Results of PPI-based triple therapy for Helicobacter pylori infection treatment.

Authors (Year, Country)	Treatment Status	N	Eradication Rate		Adverse Events	Adherence
			ITT	PP		
PPI-Clarithromycin-Metronidazole (PC1M1)						
Tepes et al. (2021, Slovenia) [24]	First-line	35	83%	83%	N/A	N/A
Long et al. (2018, China) [28]	First-line	33	63.6%	70%	45.5%	93.9%
Sue et al. (2017, Japan) [30]	First-line	30	83.3%	82.7%	Diarrhea (6.7%) Abdominal pain (3.3%) Abdominal fullness (3.3%) General malaise (3.3%)	100%
Gisbert et al. (2015, Spain) [41]	First-line	112	57%	59%	14%	94%
Gisbert et al. (2010, Spain) [35]	First-line	50	54%	55%	10%	98%
Gisbert et al. (2005, Spain) [39]	First-line	12	58%	64%	17%	92%
Nyssen et al. (2020, Europe) [25]	First-line	228	69%	69%	23%	98%
Ono et al. (2017, Japan) [32]	First-line	10	50%	55.6%	7.7%	N/A
Ono et al. (2017, Japan) [32]	Second-line	3	33.3%	33.3%	7.7%	N/A
Segarra-Newnham et al. (2004, USA) [33]	Unknown	22	N/A	91%	N/A	N/A
Adachi et al. (2023, Japan) [20]	Unknown	8	50%	50%	12.5%	100%
PPI-Metronidazole-Tetracycline (PM1T)						
Matsushima et al. (2006, Japan) [38]	First-line	5	80%	100%	N/A	N/A
Rodriguez-torres et al. (2005, Puerto Rico) [40]	First-line	17	84%	N/A	55%	80%
Rodriguez-torres et al. (2005, Puerto Rico) [40]	Second-line	3	100%	N/A		
PPI-Clarithromycin-Levofloxacin (PC1L)						
Nyssen et al. (2020, Europe) [25]	First-line	50	80%	82%	19%	98%
Gisbert et al. (2010, Spain) [35]	Second-line	15	73%	73%	20%	100%
Nyssen et al. (2020, Europe) [25]	Second-line, failed PC1M1	17	71%	69%	16%	89.5%
Gisbert et al. (2015, Spain) [41]	Second-line, failed PC1M1	50	64%	73%	23%	88%
Nyssen et al. (2020, Europe) [25]	Second-line, failed PBTM1	3	100%	100%	50%	75%
Gisbert et al. (2015, Spain) [41]	Second-line, failed PBTM1	14	64%	64%	29%	100%
Nyssen et al. (2020, Europe) [25]	Third-line	2	50%	50%	0%	100%
Gisbert et al. (2015, Spain) [41]	Third-line	3	33%	50%	67%	67%
Gisbert et al. (2015, Spain) [41]	Fourth-line	2	100%	100%	67%	100%
Gisbert et al. (2005, Spain) [39]	Fourth-line	2	100%	100%	50%	100%
PPI-Metronidazole-Levofloxacin (PM1L)						
Nyssen et al. (2020, Europe) [25]	Second-line, failed PC1M1	13	77%	77%	23%	100%
Nyssen et al. (2020, Europe) [25]	Second-line, failed PBTM1	4	75%	75%	0%	100%
PPI-Metronidazole-Sitafloxacin (PM1S)						
Ono et al. (2017, Japan) [32]	First-line	20	100%	100%	Severe adverse events (2.3%)	N/A
Mori et al. (2017, Japan) [31]	First-line	33	100%	N/A	Total (31.6%) Soft stool (12.3%) Diarrhea (7.0%) Dysgeusia (7.0%) Stomatitis (5.3%) Itching (5.3%) Skin rash (3.5%) Abdominal pain (1.8%) Headache (1.8%)	100%
Mori et al. (2017, Japan) [31]	Second-line	19	84.2%	N/A		
Mori et al. (2017, Japan) [31]	Third-line	5	40%	N/A		
Ono et al. (2017, Japan) [32]	Second-line	24	100%	100%	Severe adverse events (2.3%)	N/A
PPI-Clarithromycin-Rifabutin (PC1R)						
Gisbert et al. (2005, Spain) [39]	Third-line	9	11%	17%	89%	67%
Gisbert et al. (2015, Spain) [41]	Third-line	7	14%	20%	71%	71%
Gisbert et al. (2015, Spain) [41]	Fourth-line	2	50%	0%	100%	100%
PPI-Metronidazole-Minocycline (PM1M2)						
Osumi et al. (2017, Japan) [29]	Unknown	5	100%	N/A	N/A	N/A

Abbreviation: N, number of patients; PP; per protocol analysis; ITT, intention to treat analysis; PC1M1, PPI-Clarithromycin-Metronidazole; PBTM1, PPI-Bismuth-Tetracycline-Metronidazole; N/A, not available.

### 3.4. Bismuth-Based Therapy 

#### 3.4.1. Traditional PPI-Bismuth-Tetracycline-Metronidazole (PBTM1)

Efficacy:

First-line therapy: Two retrospective studies reported eradication rates of 92% to 94.5% with PBTM1 as a first-line therapy [25,26]. However, this study conducted in Spain did not provide detailed information regarding the dosage and duration [25]. Gao et al. [26] used colloidal bismuth pectin 150 mg three times daily, together with PPI, tetracycline, and metronidazole for 14 days, and the eradication rate was 94.5%. In contrast, in a prospective study, Gisbert et al. [41] reported an eradication rate of 75% using omeprazole, bismuth subcitrate 120 mg four times daily, and oxytetracycline or doxycycline instead of tetracycline for 10 days.

Second-line or later-line therapy: Several studies reported the eradication rates with bismuth-based quadruple therapy for patients who failed PC1M1 [25,39,41]. Seven-days of ranitidine bismuth citrate 400 mg twice daily, metronidazole 250 mg four times daily, and tetracycline 500 mg four times daily achieved a low eradication rate of 53% [39]. When oxytetracycline or doxycycline were used as alternatives to tetracyclines, an eradication rate of 38% was achieved [41]. However, the European Registry reported an eradication rate of 82% with PBTM1 without specifying the dosage and duration [25]. Among five patients who failed PC1L as the first-line therapy, the eradication rate was 80% [25].

In a prospective open-label randomized trial, 109 patients with penicillin allergy who failed one or more regimens were included [37]. The combination of bismuth potassium citrate, with PPI, metronidazole, and tetracycline achieved an eradication rate of 93.1%, with no significant difference between penicillin-allergic and non-allergic patients. For patients who failed PC1M1 and PC1L, PBTM1 has shown an eradication rate ranging from 82% to 100% in several studies [25,41]. In six patients who failed several therapies, five of them achieved successful treatment [25].

Safety:

First-line therapy: The incidence of reported adverse effects was between 14% and 46.7%, while adherence rates varied from 83.3% to 98% [25,26,41]. Gisbert et al. [41] used a 10-day oxytetracycline or doxycycline in combination with bismuth, metronidazole, and PPI, and reported a lower incidence of adverse events at 14%. Gao et al. [26] used a 14-day PBTM1 and found a higher incidence of mild to moderate adverse events at 46.7%. The most common adverse events reported included headache, dizziness, nausea/vomiting, abdominal discomfort, and diarrhea [26].

Second-line or Later-line therapy: The incidence of adverse effects for second-line or later-line therapies was higher than that of first-line therapy, ranging from 34% to 58% in studies with larger sample sizes [25,39,41]. Despite this, the adherence rate remained high, ranging from 87% to 95.3%. The most commonly reported adverse effects were nausea, abdominal pain, diarrhea, and heartburn [39].

#### 3.4.2. Modified Bismuth Quadruple Therapies

##### Minocycline Containing Therapy

Zhang et al. [23] conducted a randomized controlled trial to investigate the efficacy and safety of minocycline-containing bismuth quadruple therapy in newly diagnosed patients. The study found that PPI-Bismuth-Metronidazole-Minocycline (PBM1M2) achieved an eradication rate of 83.6%. The success rate of PPI-Bismuth-Levofloxacin-Minocycline (PBLM2) was 90.4% [23]. However, the incidence of adverse events was relatively high, with 33.8% for PBLM2 and 47.4% for PBM1M2. The common adverse reactions included nausea, abdominal discomfort, diarrhea, and dizziness [23]. Patients were highly adherent to the therapy [23].

##### PPI-Bismuth-Clarithromycin-Metronidazole (PBCM1)

Long et al. [28] performed a prospective, randomized, open-label trial to explore the efficacy and safety of PBCM1. Treatment-naïve patients received esomeprazole, clarithromycin, bismuth potassium citrate, and metronidazole for 14 days. The study reported a high eradication rate of 96%. However, 42.4% of patients experienced mild to moderate adverse effects such as nausea, abdominal pain, and dizziness, and 6.1% reported severe adverse events. Despite the occurrence of adverse events, 81.8% of patients maintained high adherence [28].

##### PPI-Bismuth-Rifabutin-Ciprofloxacin (PBRC2)

A study conducted in Australia enrolled 69 patients with penicillin allergy [36]. Patients who had previously failed the PPI-Clarithromycin-Metronidazole regimen were treated with PPI, bismuth subcitrate, rifabutin, and ciprofloxacin for 10 days. An eradication rate of 94.2% was observed. However, no information on adverse events or adherence was provided.

##### PPI-Bismuth-Tetracycline-Furazolidone (PBTF)

Lansoprazole, bismuth potassium citrate, tetracycline, and furazolidone were given to patients who had failed at least one regimen for 2 weeks, regardless of whether they had penicillin allergy [37]. PBTF achieved a high eradication rate of 96.1%, with no difference observed between penicillin-allergic and non-allergic patients; 17.6% of patients experienced moderate to severe adverse events such as fatigue, nausea, and anorexia. The adherence rate was 95.4%.

**Table 4 antibiotics-12-00737-t004:** Results of Bismuth quadruple therapy for Helicobacter pylori infection treatment.

Authors (Year and Country)	Treatment Status	N	Eradication Rate		Adverse Events	Adherence
			ITT	PP		
Classic PPI-Bismuth-Tetracycline-Metronidazole (PBTM1)						
Nyssen et al. (2020, Europe) [25]	First-line	228	91%	92%	29%	96%
Gao et al. (2019, China) [26]	First-line	112	86.7%	94.5%	46.7%	83.3%
Gisbert et al. (2015, Spain) [41]	First-line	50	74%	75%	14%	98%
Nyssen et al. (2020, Europe) [25]	Second-line, failed PC1M1	64	78%	82%	34%	95.30%
Nyssen et al. (2020, Europe) [25]	Second-line, failed PC1L	5	80%	80%	20%	100%
Gisbert et al. (2015, Spain) [41]	Second-line	24	37%	38%	58%	87%
Liang et al. (2013, China) [37]	Second- or later-line	Unknown	87.9%	93.1%	33.6%	94.4%
Gisbert et al. (2015, Spain) [41]	Third-line	3	100%	100%	67%	100%
Nyssen et al. (2020, Europe) [25]	Third-line, failed PC1M1, PC1L	12	75%	82%	58%	92%
Nyssen et al. (2020, Europe) [25]	Third-line, failed PC1M1, PM1L	5	100%	100%	0%	100%
Nyssen et al. (2020, Europe) [25]	Third-line, failed PC1L, PBTM1	1	0%	0%	0%	100%
Modified bismuth quadruple therapy						
PPI-Bismuth-Levofloxacin-Minocycline (PBLM2)						
Zhang et al. (2022, China) [23]	First-line	74	89.2%	90.4%	Total (33.8%), Nausea (4.1%) Abdominal discomfort (12.2%) Dizziness (20.3%) Diarrhea (5.4%)	≥90%
PPI-Bismuth-Metronidazole-Minocycline (PBM1M2)						
Zhang et al. (2022, China) [23]	First-line	76	80.3%	83.6%	Total (47.4%) Nausea (15.8%); Abdominal discomfort (32.9%); Dizziness (23.7%); Diarrhea (10.5%)	≥90%
PPI-Bismuth-Clarithromycin-Metronidazole (PBC1M1)						
Long et al. (2018, China) [28]	First-line	33	84.8%	96%	48.5%	81.8%
PPI-Bismuth-Rifabutin-Ciprofloxacin (PBRC2)						
Tay et al. (2012, Australia) [36]	Second- or later-line	69	N/A	94.2%	N/A	N/A
PPI-Bismuth-Tetracycline-Furazolidone (PBTF)						
Liang et al. (2013, China) [37]	Second- or later-line	Unknown	91.7%	96.1%	17.6%	95.4%

Abbreviation: N, number of patients; PP, per protocol analysis; ITT, intention to treat analysis; PM1L, PPI-Metronidazole-Levofloxacin; PC1L, PPI-Clarithromycin-Levofloxacin; PC1M1, PPI-Clarithromycin-Metronidazole; PBTM1, PPI-Bismuth-Tetracycline-Metronidazole; N/A, not available.

## 4. Discussion and Recommendations

Even though 10% of the population have reported allergies to penicillin, only 5% of those were considered to have true allergies, allowing 95% of the population with reported allergy to tolerate amoxicillin [43]. For those with non-allergic symptoms including a runny nose, diarrhea, or vomiting, it is safe to administer oral amoxicillin without a skin test [44]. However, for those with severe reactions, a penicillin skin test should be performed to confirm an IgE-related allergy prior to prescribing amoxicillin [45]. Studies have found the efficacy of levofloxacin, omeprazole, nitazoxanide, and doxycycline (LOAD) regimens ranging from 82.7 to 88.9% in patients without penicillin allergy [46,47,48,49]. The eradication rate of bismuth quadruple therapy was 81.3% in patients with *H. pylori* without penicillin allergy [50].

In the general population with *H. pylori* infection, studies have demonstrated that vonoprazan-based therapy was often more effective than PPI-based therapy [19,51,52]. In patients with penicillin allergy, Vonoprazan-Clarithromycin-Metronidazole achieved a high eradication rate of 90%, with high adherence and tolerable adverse events in treatment-naïve patients [20,30,32]. Therefore, in regions where vonoprazan is available, vonoprazan-based therapy may be considered as a first-line therapy in patients with penicillin allergies. However, additional research is needed to support its use as a second-line therapy in patients with failure to initial treatment regimen due to limited evidence from the current literature.

The efficacy of PPI-Clarithromycin-Metronidazole was suboptimal, with an eradication rate ranging from 60% to 80%, depending on metronidazole doses and duration of therapy [24,25,28,32,35,39,41]. Even if a higher dose of metronidazole and prolonged duration could overcome metronidazole resistance and elevate eradication rates, the risk of adverse events increased accordingly [53], especially nausea and taste distortion, which may be intolerable for certain patients. Therefore, it may not be used as the initial treatment in patients with penicillin allergies. There is not enough evidence to support its use as a second-line therapy.

Resistance rate to tetracycline was less than 10% in most regions [54]. Vonoprazan-Tetracycline has demonstrated promising effectiveness for patients with penicillin allergy based on results from a real-world study [21]. The efficacy of PPI-Metronidazole-Tetracycline was approximately 80% as the first-line therapy [38,40]. However, 27.8% to 50% of patients reported adverse effects [21,40]. It can be used as an alternative when other first-line therapies fail or are unavailable. Clinicians should carefully monitor patients for potential adverse events and modify treatment plans as needed if it is used in clinical settings.

Levofloxacin-based therapy, including PPI-Clarithromycin-Levofloxacin and PPI-Metronidazole-Levofloxacin, achieved an eradication rate of approximately 80% when used as a first-line regimen, and approximately 70% as a second-line therapy [25,35,41]; 16% to 29% of patients experienced mild to moderate adverse events, and high adherence was observed [25,35,41]. Therefore, levofloxacin-based treatments can be used when other treatments are not effective or are unavailable [18,55].

Despite a sitafloxacin resistance rate ranging from 21.7% to 60.3% [56], Vonoprazan-Sitafloxacin-Metronidazole achieved a high eradication rate of approximately 90% [20,22,32]. Moreover, PPI-Metronidazole-Sitafloxacin achieved an eradication rate of 100% in treatment-naïve patients, and the eradication rate ranged from 84.5% to 100% in patients who failed at least one regimen [31,32]. This phenomenon stems from the synergistic activity of metronidazole and sitafloxacin [57]. However, the incidence of adverse events of sitafloxacin-based regimens were higher than clarithromycin and metronidazole-containing regimens. More importantly, potentially permanent damage involving the skeletal and muscular system and the central nervous system were associated with fluoroquinolone use [58]. Therefore, sitafloxacin-based treatments may be used only when other treatment regimens are ineffective or unavailable.

On the other hand, the eradication rate of PPI-Clarithromycin-Rifabutin was as low as 20% when it was used as a third-line or fourth-line therapy. Moreover, the incidence of adverse effects was over 70% [39,41]. Therefore, rifabutin-based therapy is not recommended for patients with multiple treatment failures.

The eradication rate of traditional bismuth quadruple therapy with standard dose was around 90% when it was used as a first-line therapy [25,26]. A lower bismuth dose and use of oxytetracycline or doxycycline in place of tetracycline decreased the eradication rate [41]. The eradication rate of traditional bismuth quadruple therapy ranged from 82% to 100% as second-line or third-line therapy [25,37,41]. Although this traditional bismuth quadruple therapy was effective for the treatment of *H. pylori* infection, the complicated administration schedule, high incidence of adverse reactions, and unavailability of tetracycline in many regions make this combination therapy a less preferable regimen [59,60].

Some studies have modified the traditional bismuth quadruple therapy and tried different combinations of antibiotics with PPI and bismuth [23,28,36,37]. Minocycline containing bismuth quadruple therapy achieved an eradication rate of 90.4% with levofloxacin and 83.6% with metronidazole [23]. The resistance to minocycline was low, and minocycline may be a good alternative to tetracycline [61]. PPI-Bismuth-Clarithromycin-Metronidazole achieved an eradication rate of 96% [28]. The eradication rate of PPI-Bismuth-Rifabutin-Ciprofloxacin combination was 94.2% [36]. PPI-Bismuth-Tetracycline-Furazolidone achieved an eradication rate of 96.1% [37]. In general, fewer adverse events and high medication adherence were observed in these modified bismuth-based regimens. Therefore, these may be promising regimens due to excellent eradication rate, high medication adherence, fewer adverse reactions, less complicated schedule, and greater availability. Additional research is needed to determine an optimized dose, frequency, and duration of therapy.

Another approach to treat *H. pylori* infection in patients with penicillin allergy is to substitute amoxicillin with another β-lactam antibiotic, as the risk of cross-reactivity is relatively low [62,63]. Cefuroxime, levofloxacin, esomeprazole, and bismuth potassium citrate were used in patients with penicillin allergy for 14 days, and this combination achieved an eradication rate of 90.1% [27]. Mild to moderate adverse reactions were reported in 19.3% of patients, with fatigue, anorexia, abdominal discomfort, nausea, and diarrhea being the most common events; 95.3% of patients had high adherence. The use of cefuroxime offers a new alternative for patients who cannot tolerate amoxicillin, which achieves treatment effectiveness and reduces the unnecessary use of other antibiotics.

Systematic reviews have suggested that susceptibility-guided treatment may be more efficacious than empirical first-line triple therapy [64,65]. In patients with penicillin allergy, susceptibility-guided therapy achieved a high eradication rate of greater than 90% [66]. Therefore, susceptibility-guided therapy may be utilized in patients with penicillin allergy to select appropriate antibiotics.

## 5. Conclusions

In patients with penicillin allergy, Vonoprazan-Clarithromycin-Metronidazole, traditional bismuth quadruple therapy, and modified bismuth quadruple therapy have demonstrated an excellent eradication rate and high adherence. Vonoprazan-based therapy is administered less frequently and seems to be better tolerated than bismuth quadruple therapy. Therefore, vonoprazan-based therapy may be considered as a first-line therapy if accessible. Bismuth quadruple therapy can be used as the initial therapy when vonoprazan is unavailable. PPI-Clarithromycin-Metronidazole, levofloxacin-based, sitafloxacin-based regimen, and cephalosporins can be used as alternatives when first-line therapies are ineffective, intolerable, or unavailable. Susceptibility-guided therapy can assist in selecting appropriate antibiotics. PPI-Clarithromycin-Rifabutin should not be used because of its low eradication rate and frequent adverse reactions.

## Data Availability

Data sharing not applicable.

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
