# Peer review of "Treatment of Helicobacter pylori Infection in Patients with Penicillin Allergy"

_antibiotics, 2023, doi:10.3390/antibiotics12040737_

Round 1

Reviewer 1 Report

This article deals with an interesting topic concerning the treatment of Helicobacter pylori infections. It is focused on those patients allergic to penicillins for whom the use of amoxicillin which is the antibiotic of choice for this disease, is not recommended. The authors checked and reported the results of 22 articles from the literature where different treatment options were examined and evaluated based on their efficacy, safety and side effects. Vonoprazan-based therapy and Bismuth Quadruple therapy result as being the most efficacious therapies. The article is well written, well organized by examining in detail and critically the different treatment options so that it can be useful for the readers. Just a few minor  clarifications should be made by the authors. In the abbreviations on page 6, it would be appropriate to differentiate in the acronyms the antbiotics Mynocycline from Metronidazole  both indicated with the letter M and Clarrithomycin from Ciprofloxacin  both indicated with the letter C.. In the lines 161-166 contrasting results are reported. For instance MET for 14 days reached an eradication rate of 70% whereas MET at the same concentration for 7 days achieved an eradication rate of 90%, however it is well known that the more effective therapy should last 14 days. This is unclear. Further in the lines 220-222 when it refers to the falure of bismuth quadruple therapy (BQT), which BQT (PBTM or PCML) is taken into account? Lastly how is considered, for the purpose of using therapy with MZ, the fact that the high concentration of MZ leads to more side effects?

Author Response

1. In the abbreviations on page 6, it would be appropriate to differentiate in the acronyms the antibiotics Minocycline from Metronidazole both indicated with the letter M and Clarithromycin from Ciprofloxacin both indicated with the letter C.

We apologize for any confusion caused earlier. We have revised the manuscript and made the necessary changes. In the updated manuscript, we have used the following acronyms: M1 for metronidazole, M2 for minocycline, C1 for clarithromycin, and C2 for ciprofloxacin.

2. In the lines 161-166 contrasting results are reported. For instance, MET for 14 days reached an eradication rate of 70% whereas MET at the same concentration for 7 days achieved an eradication rate of 90%, however it is well known that the more effective therapy should last 14 days. This is unclear.

Thank you for your feedback. We have added a statement for clarification. Metronidazole is a concentration-dependent antibiotic; thus, higher individual doses of 750 mg may increase the eradication rate (Lines 166-167).  

3. Further in lines 220-222 when it refers to the failure of bismuth quadruple therapy (BQT), which BQT (PBTM or PCML) is taken into account?

Thank you for your comment. We have made the necessary changes in the manuscript by indicating PBTM1 as the bismuth quadruple therapy (Line 225).

4. Lastly, how is considered, for the purpose of using therapy with MZ, the fact that the high concentration of MZ leads to more side effects?

Thank you for your suggestion. Based on our research, we have found that while a higher dose of MZ may increase the eradication rate, it may also lead to a higher occurrence of adverse effects. Thus, we do not recommend using it as the first-line treatment (Lines 358-360).

Reviewer 2 Report

The manuscript is an almoust complete description of eradication possibilities of H. pylori in penicillin allergic patients. However, I have some comments concerning the content and formatting of the manuscript.

1.The statement that there are 30 millions penicillin allergic patients in USA is probably wrong. Ther prevalence s is 2-10% in patients charts.

2. In some sentences, wording is difficult to understand for a non-native English reader

3. H. pylori is not associated with GERD, on the contrary, it protects against reflux by reducing acid secretion.

4. Cefuroxim-based regimen is mentioned in the abstract but not in the main body and reference list. Nitazoxanide is included in the reference list but not commented in the text.

5. The tables are not uniform,  reference numbers. are lacking

6. The authors does not express their position regarding the need of penicillin allergy testing before prescibing any regimen, or to they choose the regimens absed on self reporting?

Author Response

1.The statement that there are 30 million penicillin allergic patients in the USA is probably wrong. The prevalence s is 2-10% in patients charts.

Thank you for your comment. We have revised the prevalence of penicillin allergy to be 4% to 15% based on the references.

2. In some sentences, wording is difficult to understand for a non-native English reader

Thank you for your feedback. We apologize for any confusion caused by the language used in some of the sentences. We have reviewed and revised the manuscript again to ensure that it is clear and easily understandable for all readers, including non-native English speakers.

If you have any specific suggestions or concerns, please feel free to let us know, and we will do our best to address them.

3. H. pylori is not associated with GERD, on the contrary, it protects against reflux by reducing acid secretion.

We appreciated your comment. We have deleted GERD in this manuscript. 

4. Cefuroxime-based regimen is mentioned in the abstract but not in the main body and reference list. Nitazoxanide is included in the reference list but not commented in the text.

Thank you for your suggestion. Cefuroxime is listed in Table 1, and it is mentioned in the manuscript (Lines 415-422). Nitazoxanide has not been studied in penicillin-allergy population, therefore, it is only briefly mentioned in the discussion (Lines 344-346).

5. The tables are not uniform, reference numbers. are lacking.

The reference numbers have been added to the Tables.

6. The authors do not express their position regarding the need of penicillin allergy testing before prescribing any regimen, or to they choose the regimens based on self-reporting?

If the patient reported a mild non-allergic symptom, amoxicillin could be given without a skin test. In patients who reported severe reactions, a skin test should be performed to confirm the true allergy prior to prescribing amoxicillin (Lines 339-342).